# Astrocyte Signaling in the Neurovascular Unit After Central Nervous System Injury

**DOI:** 10.3390/ijms20020282

**Published:** 2019-01-11

**Authors:** Lena Huang, Yoshihiko Nakamura, Eng H. Lo, Kazuhide Hayakawa

**Affiliations:** Neuroprotection Research Laboratory, Departments of Radiology and Neurology, Massachusetts General Hospital and Harvard Medical School, Charlestown, MA 02129, USA; LHUANG23@mgh.harvard.edu (L.H.); YNAKAMURA2@mgh.harvard.edu (Y.N.); Lo@helix.mgh.harvard.edu (E.H.L.)

**Keywords:** astrocyte, neurovascular unit, central nervous system

## Abstract

Astrocytes comprise the major non-neuronal cell population in the mammalian neurovascular unit. Traditionally, astrocytes are known to play broad roles in central nervous system (CNS) homeostasis, including the management of extracellular ion balance and pH, regulation of neurotransmission, and control of cerebral blood flow and metabolism. After CNS injury, cell–cell signaling between neuronal, glial, and vascular cells contribute to repair and recovery in the neurovascular unit. In this mini-review, we propose the idea that astrocytes play a central role in organizing these signals. During CNS recovery, reactive astrocytes communicate with almost all CNS cells and peripheral progenitors, resulting in the promotion of neurogenesis and angiogenesis, regulation of inflammatory response, and modulation of stem/progenitor response. Reciprocally, changes in neurons and vascular components of the remodeling brain should also influence astrocyte signaling. Therefore, understanding the complex and interdependent signaling pathways of reactive astrocytes after CNS injury may reveal fundamental mechanisms and targets for re-integrating the neurovascular unit and augmenting brain recovery.

## 1. Introduction

The neurovascular unit is a conceptual framework for investigating the mechanisms of central nervous system (CNS) injury and disease [1]. Under normal condition, crosstalk between various types of brain cells including neurons, glia, brain endothelial cells, pericytes, and stem/progenitors take place to maintain CNS homeostasis. However, after CNS injury, disruption of these cell–cell signaling results in the development of acute injury in addition to neuronal damage itself [1,2,3,4]. On the other hand, restoring the neurovascular function may lead to [5,6,7,8,9] the generation of new blood vessels and facilitating highly coupled neurorestorative processes including neurogenesis and synaptogenesis in the late phase of injury [5,6,7]. Collectively, multi-cellular processes within the neurovascular unit can mediate both damage and repair after CNS injury.

Of cells in the neurovascular unit, astrocytes comprise 19% to 40% of glial cells in the human brain including neocortical gray matter and white matter [8]. Astrocytes are highly heterogeneous and pleomorphic, and there are at least two different populations: Protoplasmic astrocytes (gray matter, type-1) and fibrous astrocytes (white matter, type-2). Protoplasmic astrocytes widely distributing in gray matter have a larger size (~50 μm) and more organelles than fibrous astrocytes, and at least one process contacts blood vessels through perivascular endfeet as well as forming multiple contacts with neurons. These astrocytes regulate local blood flow and neuronal homeostasis [9,10]. Fibrous astrocytes are originated from radial glial cells that are capable of differentiating neurons, astrocytes and oligodendrocytes during brain development, and these astrocytes highly express glial fibrillary acidic protein (GFAP), nestin, and vimentin [11,12]. Although the specific function remains to be characterized, fibrous astrocytes also contact vessel capillaries like the protoplasmic astrocytes [13]. Generally, elevated GFAP is a common feature of the activation state of astrocytes.

In a normal brain, astrocytes play key roles in sustaining CNS homeostasis. Not surprisingly, astrocytes are now known to also play a vital role in how the brain responds to injury and disease. After CNS injury, reactive astrocytes are traditionally thought to impede dendritic and axonal growth by producing inhibitory molecules [14,15,16]. However, accumulating evidence now reveal that astrocytes have a wide variety of essential phenotypes, with beneficial and deleterious actions spanning a complex spectrum from the acute stage of injury/disease to the later phases of recovery. The reader is referred to many detailed reviews that rigorously describe these multiphasic astrocyte phenotypes and functions [15,17,18,19,20,21,22,23,24]. Here, in this mini-review, we aim to survey primary principles to support the idea that astrocytes comprise a critical source of crosstalk signaling within the neurovascular unit as the damaged brain begins to transition from initial injury into endogenous repair and recovery after CNS injury, including stroke, white matter injury, and spinal cord injury.

## 2. Role of Astrocytes for Neuroprotection and Neurorepair after Stroke

### 2.1. Metabolic Connection between Astrocytes and Neurons

Under normal, homeostatic conditions, neurons preferably metabolize glucose through an oxidative branch called the pentose phosphate pathway (PPP), and produce nicotinamide adenine dinucleotide phosphate (NADPH) in this pathway together with glutathione reductase regenerating reduced glutathione, which scavenges reactive oxygen species (ROS) in the neurons [23,25,26], thus maintaining antioxidant status. In conditions where blood flow is occluded and there is a lack of supply of glucose and oxygen, there is a disturbance in the capabilities of neurons to metabolize existing glucose molecules and as a result will cause irreversible injury along with oxidative stress and apoptotic death in neurons in the ischemic core [22,23,27]. Astrocytes, on the other hand, have a high glycolytic profile and preferably metabolize glucose to yield pyruvate that can be catalyzed by lactate dehydrogenase 5 to lactate which gets shuttled to neurons for further biochemical pathways for energy synthesis. Since astrocytes contain massive glycogen storage, astrocytes can temporarily shift from oxidative phosphorylation to glycolysis [28] and metabolize existing glycogen into biomolecules essential for survival. This switch in metabolic pathway may allow astrocytes to endure prolonged periods of hypoxia, which has been seen in both in vitro [29,30,31] and in vivo [32] ischemia models. Additionally, neuronal survival may be increased by astrocyte lactate shuttling. It has reported that enhancing the lactate production in astrocytes by ischemic preconditioning treatment increased its neuroprotective ability in vitro [33]. Lactate is converted to pyruvate by lactate dehydrogenase 1 in neurons and utilized in mitochondria for adenosine triphosphate (ATP) production [34,35]; therefore, healthy mitochondria along with sufficient oxygen is essential for neurons. Although lactate-mediated neuroprotection has been documented [36,37,38], how lactate fulfills neuronal energy demand when mitochondria are dysregulated under ischemia remains to be fully elucidated.

The ability for neurons to endure prolonged periods of hypoxia was contingent upon the availability of glycogen. Brown et al. [39] also demonstrated that increasing astrocytic glycogen stores preserves the neuronal function and viability of both astrocytes and neurons under conditions of limited available energy. In addition to providing metabolic fuel, Zhao et al. [22] identified that astrocytes released ATP after stroke, which increased extracellular adenosine. Adenosine binds to A1 receptors, which subsequently inhibits adenylyl cyclase and ultimately inhibits the pathway for neuronal cell death [40,41]. This suggests that the energy provided by astrocytes not only supports metabolic energy to neurons but also may provide neuroprotective signaling.

### 2.2. Astrocytic Mitochondrial Membrane Potential and Neuroprotection

To preserve and prolong the functional capacities of astrocytic bioenergetic mechanisms and neuronal viability, considerations on how to maintain astrocyte mitochondrial membrane potential has been another area of focus. Many neuronal processes necessary for survival may partly be dependent on astrocytic mitochondrial function and associated buffering capacities. This includes maintenance of extracellular pH and potassium, Ca^2+^ buffering, and transfer of lactate and/or pyruvate to neurons as energy substrates [42,43,44,45]. It is postulated that the astrocytic mitochondria undergo early depolarization during oxygen-glucose deprivation (OGD) to allow a shift away from aerobic metabolism to glycolysis, which supplies energetically impaired neurons with lactate [46] in order to spare neurons from death. One potential treatment for preventing astrocyte mitochondrial membrane depolarization is to inhibit nitric oxide synthase in astrocytes after OGD [28]. Utilization of this inhibitor suggests that nitric oxide (NO) or peroxynitrite are involved in the loss of mitochondrial membrane potential, presumably through the inhibition of electron transport chain activity [47,48]. Although this may suggest that inhibition of nitric oxide synthase (NOS) should rescue astrocytic mitochondrial membrane potential, a caveat to consider is that NOS activity is present not only in the astrocyte but also globally, and so non-targeted NOS inhibition may cause off-target side effects.

Moreover, selective purinergic agonist, 2meSADP, enhances mitochondrial ATP production in astrocytes in an IP_3_ receptor-dependent manner and reduces neuronal cell death after transient focal ischemia in rats [49]. In contrast, inhibition of astrocytic mitochondria with fluorocitrate makes neurons vulnerable to cytotoxicity [50] and retards neurovascular remodeling and functional recovery after stroke [51], suggesting that mitochondria-targeted therapy to restore or enhance their function in astrocytes may augment neuronal survival even during the recovery period after stroke.

### 2.3. Astrocytic Mitochondria for Neuroprotection and Recovery

Astrocytes play broad roles in the CNS, but perhaps most importantly, they protect neurons against oxidative stress and excitotoxicity [52]. Healthy mitochondria are essential for these neuroglial protective mechanisms due to the fact that inhibition of astrocytic mitochondria makes neurons vulnerable to cell death [50]. Recently, intercellular mitochondrial transfer has been proposed as a new paradigm for cell–cell signaling [53,54]. In the CNS, it has been reported that retinal neurons may transfer mitochondria to astrocytes for processes such as disposal and recycling [55]. Microglia may exchange mitochondria with adjacent neurons for neuroprotection [56]. A recent study proposed that, under some experimental conditions, astrocytes may transfer mitochondria to neurons and promote neuroprotection and improve neuroplasticity after stroke [57]. This ability to exchange mitochondria may signify a potential mode of cell–cell signaling in the CNS.

Finally, extracellular mitochondria may also become biomarkers to indirectly evaluate the CNS metabolic integrity. Assessment of extracellular mitochondria by flow cytometry analysis suggested that human astrocyte-derived extracellular mitochondria retained higher membrane potential compared to mitochondria from human brain endothelial cells or human pericytes [58]. Proof-of-concept study was recently performed in a small cohort of subarachnoid hemorrhage (SAH) patients [59]. After SAH, an increase of astrocyte-derived mitochondria and a higher mitochondrial membrane potential at day 3 post-SAH were correlated with good clinical recovery at 3 months. Collectively, although further careful investigation is required, an assessment of extracellular mitochondrial function in cerebrospinal fluid (CSF) may enable the monitoring of brain metabolic status during therapy-utilizing mechanisms of intercellular mitochondrial transfer as well as viable mitochondrial transplantation after CNS injury.

## 3. Astrocytic Factors Regulate Vascular Function and Remodeling after CNS Injury

### 3.1. Nitric Oxide in Vascular Inflammation and Lactate Shuttle

Nitric oxide plays an important role in vascular regulation; therefore, loss of vasoregulatory effects of endothelial NO may lead to vasoconstriction and subsequently have a negative impact on the regulation of microvascular flow [60]. Additionally, inhibition of NO would cause a loss of anti-aggregant, anti-proliferative, and anti-cell adhesion effects of NO, which would subsequently promote vascular inflammation including promotion of platelet aggregation, leukocyte adhesion to endothelial cells, and smooth muscle proliferation [61].

It is suggested that endothelial NO plays a role in astrocyte-neuron coupling via the lactate pool [62] after increased blood flow. Nitric oxide can induce glycolytic flux in astrocytes by blocking complex IV, which subsequently inhibits mitochondrial respiration and spares glycogen from being metabolized into glucose [63]. San Martin [64] suggested that NO contributes to cellular cross-talk in the brain during increased blood flow because inhibition of complex IV preserves O_2_ for further use by neurons and allows for induction of glycolytic flux and lactate production in astrocytes, forming an extracellular lactate pool that will subsequently supply neurons.

### 3.2. Astrocytic Involvement in Vascular Remodeling after Stroke

During brain ischemia, there is a series of events that may compromise the integrity of the blood–brain barrier (BBB). The deprivation of glucose and oxygen triggers cellular stress in the neurovascular unit (NVU), which includes the vascular system, glial cells and neurons. The cellular stress causes release of oxygen free radicals. Subsequently, there is production of pro-inflammatory cytokines that allow for increased permeability of the BBB and therefore infiltration of leukocytes. This may exacerbate the inflammatory response and increase apoptotic signaling [65,66]. Astrocytes may respond through the release of a number of extracellular factors such as Erythropoietin (EPO), vascular endothelial growth factor (VEGF), and glial derived neurotrophic factor which all could reduce neurovascular damage and improve neuronal function after CNS injury [67,68,69].

For many years, the CNS was considered an immune-privileged organ. However, emerging data suggest that crosstalk between the brain and systemic responses is also important in both injury and repair after CNS injury [60]. For example, circulating endothelial progenitor cells (EPCs) have been documented as critical elements in tissue vascularization and endothelium homeostasis after CNS damage and neurodegeneration [70,71]. Endothelial progenitor cells are able to migrate to injured tissue and induce angiogenesis, vasculogenesis and neurovascular tissue repair [72,73]. We found that endogenous high-mobility group box 1 (HMGB1) produced by astrocytes was crucial for EPC homing in vivo in both gray matter and white matter [74,75]. In vitro, we confirmed that reactive astrocytes released HMGB1 that upregulated receptor for advanced glycation end products (RAGE) on brain endothelial cells, thus augmenting β2-integrin-mediated EPC adhesion and transmigration [76]. Ultimately, how EPC-derived signals regulate vascular remodeling requires further examination.

### 3.3. Vascular Remodeling and Oligodendrogenesis after White Matter Injury

In white matter, the neurovascular unit is even more complex, with additional cell signaling from oligodendrocytes and their precursors. It is now recognized that in white matter, astrocytes, oligodendrocyte precursor cells and cerebral endothelial cells all work together to maintain the BBB function [77]. White-matter damage is a clinically important focus for cerebrovascular disease due to its high sensitivity and vulnerability to ischemic and oxidative stress [1], and disruption of crosstalk between cells in white matter is now considered to play a critical role in vascular-basis cognitive dysfunction and dementia. In white-matter injury, reactive astrocytes promoted angiogenesis through endogenous EPC homing into the injury sites [75]. More recent studies demonstrate that exogenous EPC or EPC-derived soluble factors increased the thickness of the corpus callosum, angiogenesis, and oligodendrogenesis [78,79], suggesting that the crosstalk between oligo-endothelium/EPC-astrocyte may be essential in order to accelerate white-matter remodeling [80].

## 4. Astrocytes and Immune Regulation during CNS Inflammation

Astrocytes express Toll-like receptors, nucleotide-binding oligomerization domains, double-stranded RNA-dependent protein kinase, scavenger receptors, mannose receptor and components of the complement system that are known to regulate innate immune responses. Under some conditions, activated astrocytes produce soluble factors, such as C-X-C motif chemokine 10 (CXCL10), C-C motif chemokine ligand 2 (CCL2), interleukin-6, and B-cell activating factor (BAFF), and accelerate both innate and adaptive immune responses [81]. Gene profiling of human astrocytes stimulated with poly(I:C) demonstrated the increase of an antiviral response [81]. Double-stranded (ds)RNA triggers glial activation in mice accompanied with upregulation of inducible nitric oxide synthase (iNOS) and upregulation with nitric oxide production [82] through Toll-like receptor 3 (TLR3) signaling. Indeed, injection of (ds)RNA triggers microglial and astrocytic activation in wild-type mice but not in TLR3 knockout mice [78,79].

After CNS injury, activated astrocytes may participate in inflammatory processes through crosstalk with microglia. High level of S100β production in astrocytes further contribute to iNOS and NO production in microglia [83]. Astrocytes are able to downregulate antigen-presenting function of invading monocytes [84]. Furthermore, astrocytes can also shift microglia phenotype. It has been reported that endothelial-activated microglia were neurotoxic, whereas astrocyte-activated microglia promoted neuronal dendritogenesis through upregulating beneficial phenotype markers, including insulin-like growth factor (IGF-1) and CD206 [85]. Astrocytic function is also influenced by microglial subsets. For example, lipopolysaccharide-stimulated microglia turned astrocytes into neurotoxic phenotype [86], whereas microglia can also transform astrocytes into a neuroprotective phenotype through downregulating astrocytic P2Y1 [87]. In the context of CNS repair after injury, how the crosstalk between astrocytes and microglia regulates detrimental versus beneficial effects should be further investigated.

## 5. Astrocytes and Stem/Progenitor Cells during CNS Recovery

### 5.1. Astrocyte Regulation of Stem/Progenitor Cell Response in the Brain and Spinal Cord

The use of stem cells for neuronal recovery and regeneration after ischemia have become a growing interest as a potential form of therapy. Endogenous stem cells have been found in the subventricular zone (SVZ), hippocampus, and central canal of the spinal cord [88,89,90]. Another progenitor cell identified was GFAP-expressing, which showed increased induction of neurogenesis after stroke. These GFAP-expressing progenitor cells were found in the SVZ and migrated newly born neurons into a unique neurovascular niche in peri-infarct cortex [6]. The potential to manipulate these progenitor cells to induce changes in stem cells provide promise. For instance, an in vivo study utilizing forced expression of *Ascl1* of SVC endogenous stem cells showed sufficient conversion into neurons both in vitro and in vivo and may be effective as a target for neuron repair [91].

Studies have also shown that ischemic insult can influence astrocytes to express proliferative capabilities and acquire stem-cell hallmarks under in-vitro conditions. Doetsch et al. [92] demonstrated that, in the SVZ of mouse brain, a subpopulation of astrocytes could give rise to immature precursors and neuroblasts both in vivo and in vitro. In another study done by Jiang et al. [93], Olig2PC-astroglial cells were identified as one of the types of astrocyte cells that contained stem-cell qualities. When grafted onto brains that were subjected to global ischemia, Olig2PC-Astros exhibited neuroprotective effects and showed improvement in behavior.

To cater to the issue of limited available endogenous neural stem cells, exogenous neuronal stem cells have been considered as a potential solution. Utilization of embryonic stem cells and induced pluripotent stem cells (iPS cells) transplanted to the damaged CNS tissue [94,95] showed modest recovery in the areas affected by CNS injury. Another method of exogenous stem-cell therapy proposed is the use of bone marrow-derived mesenchymal stromal cells. This therapy contains a population of cells with both progenitor and stem cells that have shown improved functional recovery in rodent stroke models when they were delivered several hours to days after the onset of stroke [96,97]. The administration of this therapy showed an increase in axonal density, a reorienting of axonal growth, and an increase in transcollosal branching of axons into the contralateral cortex and contralateral sprouting of axons in the spinal cord [98]. The mechanism for this effect is suggested by induction of neurotrophic factor expression and release from astrocytes including VEGF, brain-derived neurotrophic factor (BDNF), and basic fibroblast growth factor (bFGF) [99]. Additionally, the use of mesenchymal stromal cells may diminish glial scar formation after stroke [22]. Although this therapy may provide promising results for neuroregeneration, we must recognize that glial scar formation serves as a double-edged sword in neuro-recovery after stroke. As the use of mesenchymal stromal cell therapy may diminish glial scar formation, this may also prevent functional benefits of glial scarring. In the next sections, we will re-evaluate past dogma regarding the harmful qualities of glial scars and suggest that glial scars may contribute to neuroregeneration.

### 5.2. Reprogramming Reactive Astrocytes into Neurons

Sanai et al. [100] showed that SVZ astrocytes in the adult human brain are clonal precursors of self-renewing multipotent neurospheres and can be differentiated into neurons in the absence of exogenous growth factors. The potential to reprogram astrocytes into neurons has become a growing interest, specifically with the focus on how to exploit this mechanism as an effective therapy for neuroregeneration. Since then, several studies have utilized different methods to redirect astrocyte expression. Heins [101] and Berninger [102] have shown that postnatal astrocytes are capable of being redirected towards neurons by forced expression of Pax6 or pro-neural transcription factor neurogenin-2 in vitro. Heinrich et al. [103] also isolated astrocytes from an adult cortex, reprogrammed the astrocyte with pCAG-Neurog2-containing retroviral vector, and was able to give rise to synapse-forming glutamatergic neurons. Neural stem cells (NSC) were also generated by reprogramming astrocytes with individual stem transcription factors octamer-binding transcription factor 4 (OCT4), sex determining region Y-Box 2 (SOX2), or Nanong homeobox (NANOG). These astrocyte-derived NSCs were able to generate mature neurons in vivo with positive expression of synaptic proteins and neurotransmitters [104]. However, shortcomings of reprogramming astrocytes were observed when further transplantation of these astrocyte-derived stem cells in vivo, which resulted in differentiation into glial cells instead of functional neurons. There was also inconsistency among the majority of astrocytes in the CNS to reprogram into neurons after injury [88]. Instead, astrocytes also formed glial scars.

As we have mentioned in previous sections, glial scars may partially serve beneficial properties that offer wound healing, limit inflammation, and protect healthy tissue and therefore may be essential for spontaneous neuronal growth. One proposed strategy for CNS regeneration, then, is to redirect astrocytes towards becoming neurons after the formation of glial scarring. Recently, Guo et al. [105] reprogrammed reactive glial cells including astrocytes and NG2 glia in the glia scar into functional neurons in the adult mouse cortex by infecting these glial cells with a retrovirus-encoded single transcription factor, NeuroD1. In results, the astrocytes were mainly reprogrammed into glutamatergic neurons. The use of human astrocytes that were immunopositive for GFAP and S100B (which are markers for astrogliosis) in culture also resulted in successful reprogramming of astrocytes into functional neurons after expressing NeuroD1. This suggests that in-vitro and in-vivo reprogramming of reactive glial cells into functional neurons after brain injury or in diseased mouse brain might represent a promising therapeutic approach for concurrently modulating reactive glial scar formation and serving as an alternative mechanism for neuroregeneration. Although this may ameliorate the original nerve injury, more studies are required to determine whether the reprogramming of reactive astrocytes into functional neurons can truly rescue the residual behavioral deficits and cognitive impairments associated with neuronal damage after ischemia. Additional consideration needs to be given to how to avoid off-target effects of using retroviral injections or if there is an alternative method for gene expression that may serve as a safer therapy.

## 6. Role of Astrocytes in Glial Scar Formation after Spinal Cord Injury

Astrocytes typically have a stellate morphology but can change to a reactive state under stress [65]. In the reactive state, astrocytes express GFAP [65,106]—also known as reactive astrogliosis. Astrogliosis tends to accumulate in the focal lesions and create glial scars, which are infamously known to express inhibitory molecules, including proteoglycans, which form an unfavorable environment for axonal outgrowth. Glial scar formation is also associated with upregulation of GFAP and other genes, hypertrophy of cell bodies and process, and interaction with other types of glial cells [15,65]. Ultimately, glial scar formation has been widely accepted as a negative feature of reactive astrocyte activity, because it has been considered to be the main culprit for the inhibition of axon regeneration.

Although astrocytes may express inhibitory molecules, Anderson et al. [107] confirmed that astrocytes were not primarily responsible for the release of chondroitin sulfate proteoglycan (CSPG), a mediator that has been implicated in inhibiting axonal regeneration. Even though CSPG can be produced by astrocytes, other cells in spinal cord injury (SCI) lesions, including pericytes, fibroblast lineage cells, and inflammatory cells [108], are known to also produce CSPG and GFAP. In SCI, for lesions of mice models that either selectively killed proliferating scar-forming astrocytes [109] or deleted signal transducer and activator of transcription 3 (STAT3) signaling selectively from astrocytes [110], CSPG areas were not significantly decreased in injured core or in the area lacking astrocyte accumulation. These findings demonstrate that other cells beside astrocytes produce substantial amounts of CSPG and that blockade of astrocytic scar fails to decrease total CSPG production in SCI lesions. Therefore, astrocyte scar formation is not the true culprit for the release of molecules that inhibit axonal regeneration.

The formation of astrocyte scarring has been deemed beneficial and necessary for isolating the injury site [111], because the physical barrier limits leukocyte extravasation and protects neurons against harmful substances that are released from the infarct core. However, this is an implication that the barrier made by the astrocytic scar can only be appreciated for its indirect role in promoting BBB repair and neuronal survival. Recent evidence, however, suggests that the barrier may have a direct impact on promoting neurogenesis. When astrocytic scars were ablated, the descending corticospinal tract (CST), ascending sensory tract (AST), or descending serotonergic (5HT) axons failed to grow spontaneously [107], suggesting that there may be an endogenous component in astrocytic scars that may directly influence the regeneration of axons. It is necessary to look further at the endogenous activity that astrocytic scars may exhibit in allowing for spontaneous neuronal growth. Due to the emerging evidence that has contradicted the central dogma underlying the role of astrocytic glial scar formation, further investigation and re-evaluation of previous findings need to be made to clarify the harmful and beneficial qualities of glial scarring. A review by Sofroniew in 2018 [112] offers further detail regarding the common misconceptions and controversies regarding the failure of effective regeneration associated with astrocyte scarring.

## 7. Conclusions

Cell–cell signaling between neuronal, glial and vascular compartments provides the signals and substrates for investigating the mechanisms of acute injury and delayed recovery after stroke, brain injury and neurodegeneration [3,113]. The development of therapies to reconnect the diverse communicating signals between multiple cell types will be challenging. During normal brain function, astrocytes are primarily responsible for ionic homeostasis, modulation of synaptic activity and neurotransmission, maintenance of the BBB, and cerebral blood flow regulation and metabolism. After CNS injury and disease, reactive astrocytes represent a complex source of intercellular signals that may impede recovery or potentially contribute to neuroprotection and defense against damaging immune response (Figure 1). The balance between deleterious and beneficial responses warrants careful study. Ultimately, a more nuanced approach may be needed in order to stimulate the beneficial astrocyte program to reconnect and regenerate the entire neurovascular unit after CNS injury or disease.

After central nervous system (CNS) injury, reactive astrocytes are capable of communicating with each cell type including microglia, endothelial cells, endothelial progenitor cells (EPCs), and neurons, and this cell–cell interaction can regulate immune response, angiogenesis, vasculogenesis, oligodendrogenesis, neuroprotection, neuroplasticity, axonal growth or inhibition depending on the context. Bi-directional arrows show the mutual interaction. Broken arrows indicate other interactions that astrocytes are indirectly involved in.

## Figures and Tables

**Figure 1 ijms-20-00282-f001:**
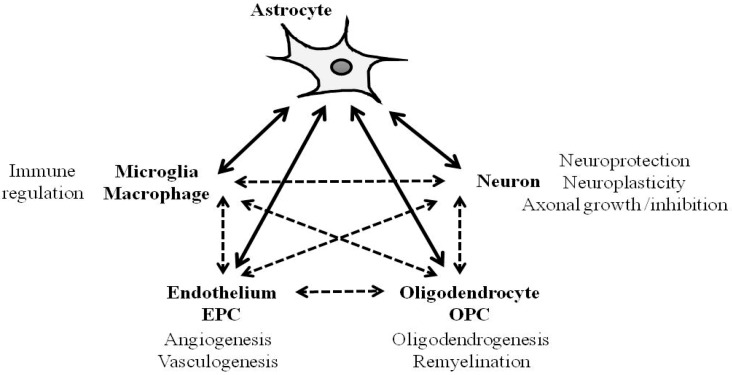
Schema of crosstalk between astrocytes and other cell type after CNS injury.

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
