# Peer review of "Astrocyte Signaling in the Neurovascular Unit After Central Nervous System Injury"

_ijms, 2019, doi:10.3390/ijms20020282_

Reviewer 1 Report

The mini-review by Huang et al. describes the influence of astrocytes on other cell types following CNS injury with a special focus on the neurovascular unit and glial scar. While this article is of potential interest to a broad scientific audience, there are a few revisions that will add clarity and improve the quality of the manuscript.

(1) While the manuscript discusses central nervous system injury, the authors primarily draw from findings from stroke models throughout the manuscript with spinal cord injury studies discussed near the end. The authors should clarify the context of each CNS injury being discussed throughout the manuscript.

(2) Although the title suggests that the review will focus on the Neurovascular Unit, the Neurovascular Unit is never clearly defined; as written, it is just a collection of neurons, glia and vascular cells. The authors should define how this collective function as a unit.

(3) The manuscript is focused on astrocytes, but it is often confusing whether the authors are referring to white matter (WM) or gray matter (GM) astrocytes in CNS injuries described. It would also help readers if the authors define the key differences between WM and GM astrocytes such as endogenous GFAP expression, morphology, role in neurovascular unit during normal conditions.

Author Response

1. While the manuscript discusses central nervous system injury, the authors primarily draw from findings from stroke models throughout the manuscript with spinal cord injury studies discussed near the end. The authors should clarify the context of each CNS injury being discussed throughout the manuscript.

Response: We appreciate for the suggestion. We now clarified the context of each CNS injury in the subtitle/heading for each section.

2. Although the title suggests that the review will focus on the Neurovascular Unit, the Neurovascular Unit is never clearly defined; as written, it is just a collection of neurons, glia and vascular cells. The authors should define how this collective function as a unit.

Response: Thank you for the suggestion. We added sentences to clarify the Neurovascular Unit in the first paragraph of the Introduction.

3. The manuscript is focused on astrocytes, but it is often confusing whether the authors are referring to white matter (WM) or gray matter (GM) astrocytes in CNS injuries described. It would also help readers if the authors define the key differences between WM and GM astrocytes such as endogenous GFAP expression, morphology, role in neurovascular unit during normal conditions.

Response: Thank you for the suggestion. We modified the Introduction with adding details to clarify astrocyte types as requested.

Reviewer 2 Report

The manuscript by Huang and colleagues presents a review of the role of astrocytes and astrocyte interactions with other CNS cells during injury. Overall, this is a brief and superficial tour of a large body of research. The reader is left with the impression that astrocytes can elicit both positive and negative effects, presumably dependent on the specific circumstances and conditions. However there is little insight or attempt to rationalise this.

Subsections and their headings are not cohesive, with several concepts scattered throughout the review, such as nitric oxide signalling and glial scarring.

The first paragraph of the Introduction section is needlessly specific and out of place with the main body of the review, with most aspects mentioned not discussed in more detail elsewhere. Thus it should be removed. A more appropriate introduction could include a very brief discussion of types of CNS injury to set the scene.

Page 2, line 13: Generally considered that astrocytes shuttle lactate to neurons as energy substrate, not pyruvate.

Page 2, line 18: Lactate is a substrate for oxygen-dependent mitochondrial oxidative phosphorylation energy production. How will lactate support neurons during hypoxia?

Page 6, line 29: Check title.

Define all abbreviations, including OGD, HMGB1, CXCl10, CCL2, BAFF, TLR3, GM-CSF.

Author Response

1. Subsections and their headings are not cohesive, with several concepts scattered throughout the review, such as nitric oxide signalling and glial scarring.

Response: Thank you for pointing this out. We tried to adjust headings to be more cohesive.

2. The first paragraph of the Introduction section is needlessly specific and out of place with the main body of the review, with most aspects mentioned not discussed in more detail elsewhere. Thus it should be removed. A more appropriate introduction could include a very brief discussion of types of CNS injury to set the scene.

Response: Thank you for the suggestion. We now modified the Introduction with deleting original sentences and adding new sentences to describe the Neurovascular Unit, types of astrocytes and types of CNS injury that we described in our manuscript.

3. Page 2, line 13: Generally considered that astrocytes shuttle lactate to neurons as energy substrate, not pyruvate.

Response: We apologize for the confusion. We now corrected the sentence.

4. Page 2, line 18: Lactate is a substrate for oxygen-dependent mitochondrial oxidative phosphorylation energy production. How will lactate support neurons during hypoxia?

Response: Thank you for reminding us this important point. We agree with the reviewer that under cerebral ischemia, the lack of oxygen may disrupt overall mitochondrial metabolic pathways. The basic hypothesis in the lactate shuttle is that transferred lactate from astrocytes to neurons is converted to pyruvate by lactate dehydrogenase 1, and newly produced pyruvate is utilized by mitochondria to produce ATP. But dysfunctional mitochondria with low oxygen environment should fail to produce ATP and fulfill neuronal energy demand even high accumulation of lactate in ischemic core (Nedargaard et al., 1991). Therefore, we believe that lactate-mediated neuroprotection may be induced depending on timing and brain area after stroke. It may be induced after reperfusion and ischemic penumbra where generally has higher blood supply with higher oxygen and more survived mitochondria compared to ischemic core. There are a couple of literature showing lactate-mediated neuroprotection after focal ischemia and reperfusion (Berthet et al., 2009, 2012, Castillo et al., 2015, Narayanan and Perez-Pinzon, 2017), but not in permanent ischemia. However, the neuroprotective mechanism under hypoxia/ischemia remains to be fully elucidated. We now included this point in our revised manuscript.

5. Page 6, line 29: Check title.

Response: We are sorry for the typo. We now changed the heading.

6. Define all abbreviations, including OGD, HMGB1, CXCl10, CCL2, BAFF, TLR3, GM-CSF.

Response: We apologize for these abbreviations. We clarified them in the revised manuscript.

Round  2

Reviewer 1 Report

The authors have satisfied all of reviewer's concerns and the manuscript is much more organized and easy to follow.  With a few minor grammatical revisions, this manuscript will be of broad interest to the research community. 

Reviewer 2 Report

All comments have been adequately addressed. The new text contains multiple minor typographical/spelling errors.